# The Correlation between Hofstede’s Cultural Dimensions and COVID-19 Data in the Early Stage of the COVID-19 Pandemic Period

**DOI:** 10.3390/healthcare11162258

**Published:** 2023-08-10

**Authors:** Ling-Hsing Chang, Sheng Wu

**Affiliations:** 1Department of Information Management, National Sun Yat-Sen University, Kaohsiung 804201, Taiwan; cchangmis@gmail.com; 2Department of Information Management, Southern Taiwan University of Science and Technology, Tainan 710301, Taiwan

**Keywords:** Hofstede’s national cultural theory, COVID-19, meta-research method, narrative method

## Abstract

COVID-19 (coronavirus disease 2019) has become the deadliest virus to affect the international community in recent history, with more than 760 million people infected and more than 6.87 million deaths as of March 2023; therefore, based on Hofstede’s national cultural theory, this study collected Hofstede’s six national cultural dimensions on a global scale, namely, power distance (PDI), individualism/collectivism (IDV), masculinity/femininity (MAS), uncertainty avoidance (UAI), long-term/short-term orientation (LTO), and indulgence/restraint (IVR) scores, and COVID-19 data from the World Health Organization (WHO) from 22 February 2020 to 30 February 2021. Then, based on eight items of global COVID-19 data, this study analyzed the correlation between Hofstede’s six dimensions and the COVID-19 data from six regions (Africa (AFRO), Europe (EURO), the Americas (AMRO), the Western Pacific (WPRO), South East Asia (SEARO), and the Eastern Mediterranean (EMRO)) divided by the WHO. This study found the following: (1) Hofstede’s six cultural dimensions indeed have a significant correlation with the COVID-19 data of different WHO regions in different ways. (2) Except for IDV and UAI, PDI is a highly critical factor and has a significant correlation with the COVID-19 data from AFRO and EMRO. MAS also is an important factor and has a significant correlation with COVID-19 data from WPRO and SEARO. Meanwhile, LTO has a significant correlation with some COVID-19 data from the AMRO region, and IVR has a significant correlation with some COVID-19 data from the EURO region. Finally, the new insights from this study are worthy of further study by scholars, and they will be of great help to global governments and medical institutions in formulating policies to suppress infectious diseases in the future.

## 1. Introduction

Since the beginning of 2020, as the coronavirus disease 2019 (COVID-19) swept the world, in order to curb the spread of this disease and reduce the threat to people’s lives, all countries issued blockade orders to lock down cities and countries and restrict the flow of all items (e.g., food, medicine), resulting in a complete change in the pace of life of people around the world. Not only was it impossible to reunite with relatives and friends during holidays, it was also impossible to go to work as usual and travel freely by means of transportation, which completely subverted people’s existing way of life [1]. In the first two decades of the 21st century, human beings have also faced many pandemic diseases (e.g., bird flu, SARS, swine flu, MARS, Ebola, Zika, and other viruses), but none were as far-reaching as COVID-19, which led to the closure of small businesses such as cafes, restaurants, and hair salons; impacted economic activity such as e-commerce, technology, and business travel; and caused the unemployment of millions of people worldwide (e.g., immigrants, part-time employees).

As of March 2023, COVID-19 has become the deadliest virus to affect the international community in recent history [2], with more than 760 million people infected and more than 6.87 million deaths. Many scholars [3,4,5,6,7,8,9,10,11,12,13,14,15,16,17] have conducted research on COVID-19 based on Hofstede’s cultural theory to study the correlation between Hofstede’s dimensions, COVID-19 data, and the role of socio-economic factors, and they found that different cultures will indeed influence COVID-19 data in different countries. In light of this, culture is an important factor influencing COVID-19 data. However, these studies did not focus on the difference between Hofstede’s cultural dimensions to explore the gap between different World Health Organization (WHO) regions.

Therefore, this research aims to further understand why different national cultures lead to different results with regard to curbing the spread of COVID-19. In view of this, based on Hofstede’s national cultural theory, this study tries to understand the correlation of COVID-19 spread in different WHO regions. Therefore, this is an important research question to be resolved in this study.

In order to clarify the research question of this study, the six dimensions of Hofstede’s national cultural theory are used. Hofstede’s theory includes countries around the world and is updated year by year, and is used as the framework for our analysis [18,19,20]. Hofstede’s national cultural theory includes the following six dimensions: (1) power distance (PDI), (2) individualism/collectivism (IDV), (3) masculinity/femininity (MAS), (4) uncertainty avoidance (UAI), (5) long-term/short-term orientation (LTO), (6) indulgence/restraint (IVR). Accordingly, through this research we hope to help the industry and government departments conduct analyses according to Hofstede’s national cultural theory [18,19,20], to reduce the spread of COVID-19, and to keep people alive in a safe IT environment.

Therefore, in this study we collect relevant secondary data (e.g., global pandemic data, related journal articles, newspapers, and magazines) and analyze these data according to Hofstede’s differences in the six dimensions of national culture based on COVID-19 data from six regions (Africa (AFRO), Europe (EURO), the Americas (AMRO), the Western Pacific (WPRO), South East Asia (SEARO), and the Eastern Mediterranean (EMRO)), as divided by the WHO. We analyze the correlation between Hofstede’s six dimensions and the COVID-19 data of six regions, and the gap between different WHO regions. Please refer to Figure A1 for the countries included in each WHO region.

However, there are five limitations of this study: (1) Because of the limitations of Hofstede’s cultural dimensions, COVID-19 data from only 117 countries can be analyzed. (2) There are some countries still at war, with no sound medical systems, and no access to the Polymerase chain reaction (PCR) test, and therefore, the research results are biased. (3) The execution of the isolation policy for each country is different, so the results of some of Hofstede’s cultural dimensions (e.g., PDI, IDV, and LTO) are conflicting and opposing. (4) The weakness of the classification of the WHO regions may be not appropriate for analyzing COVID-19 data. (5) This study does not analyze the relationship between information technology (IT) and COVID-19 data, and IT could be a critical factor in inhibiting the spread of COVID-19. However, these limitations can be a good way to direct further study.

## 2. Literature Review

At the beginning of 2020, the sudden outbreak of COVID-19 had an earth-shaking impact on industries and governments around the world. Everyone was overwhelmed for a while. After a long 10 months, the COVID-19 vaccines appeared one by one in various countries. Therefore, governments of various countries still needed to use coercive means to regulate people’s daily life in order to effectively control the COVID-19 pandemic. Although many scholars have published many papers relating to COVID-19 in the past two years, some scholars have also studied the impact of the use of IT on e-commerce. However, they mostly focus on the demographic data of users (e.g., age, gender, education level, place of residence, and income) [21], or analyze whether consumers are willing to pay for the ingredients of recipes online, so as to improve their willingness and behavior when online shopping [22].

Some scholars have analyzed the gap between different national cultures from the dimensions of individualism/collectivism and uncertainty avoidance in Hofstede’s national cultural perspective [5,13,16,23,24]. Meanwhile, Shetty et al. [14] studied an overview of five of Hofstede’s dimensions (PDI, IDV, MAS, UAI, and LTO) and their impact on the implementation of COVID-19 control strategies. Timo et al. [15] studied the correlation between Hofstede’s six cultural dimensions, COVID-19 data, and the role of socio-economic factors drawn after the pandemic. However, these studies did not focus on the difference of Hofstede’s cultural dimensions in exploring the gap between different WHO regions.

Therefore, we believe that the six dimensions of Hofstede’s national cultural theory, which includes countries around the world and is updated year by year, can be used as the framework for analysis [18,19,20] to achieve the purpose of this study. Therefore, this study will discuss the literature of Hofstede’s national cultural theory, national culture, and COVID-19.

### 2.1. Hofstede’s National Cultural Theory

Hofstede and Bond [20] believe that culture is composed of common ideas shared by a group, and is a collection of these common ideas interacting, which affects the group’s response to the environment and is different from the ideas of other groups. From 116,000 IBM employees in 72 different countries, Hofstede [18] obtained attitude scale questionnaires to understand their cultural values and ideas. A total of 53 cultural blocks were divided into four national cultural dimensions, which were analyzed, and the differences among them were compared. The participants of Hofstede’s research were employees of the same company with the same position, but they had grown up in different cultures. Therefore, the research has value is not only in its large number of participants, but also in its ability to provide researchers with an independent analysis for the discussion of cultural factors. Hofstede developed four dimensions of national cultures: power distance, individualism/collectivism, uncertainty avoidance, and masculinity/femininity.

Since the management philosophy of the East has gradually been regarded as a historical relic and a sign of backwardness, it has gradually been forgotten by the world. However, looking back on Chinese history, since Dong Zhong Shu of the Han Dynasty respected Confucianism and ousted hundreds of schools, successive dynasties have established a fairly stable social structure based on Confucianism to govern the vast territory of China, and the management structure and thought of the Chinese style should also have its contribution [20]. Therefore, Hofstede and Bond [20] obtained a new cultural dimension when conducting Chinese Value Surveys (CVSs) and analysis in their research on cross-cultural differences: Confucian dynamism. They believe that this cultural dimension is related to the tendency of long-term thinking and short-term thinking in life, and these values are related to the teachings of Confucius, so this dimension is called Confucian. Hofstede and Bond [20] surveyed 100 individuals (male: 50; female: 50) in 22 countries with this cultural dimension, and then joined China to conduct research in eight languages. Hofstede compared the research results of the CVS on IBM employees and found that IBM’s four research dimensions represented the combination of Western values and CVS represented Eastern values. Therefore, CVS was added to Hofstede’s cultural model to become the fifth dimension, which is called long-term/short-term orientation. Subsequently, a sixth dimension was added in 2010: indulgence/restraint, which is the degree to which members of society intend to control their own desires [19]. Finally, Hofstede collected data from 117 countries around the world [19].

Therefore, based on Hofstede’s cultural theory, this study discusses the six cultural dimensions proposed by Hofstede [19]: power distance, individualism/collectivism, uncertainty avoidance, masculinity/femininity, long-term/short-term orientation, and indulgence/restraint.

#### 2.1.1. Power Distance

This is the extent to which a society accepts that power is unequally distributed among its members. Hofstede [23] used the power distance index (PDI) to represent the level of social power distance. Countries with a higher level of power distance tend to centralize power and attach importance to tradition, authority, and social class. On the contrary, in countries with a lower level of power distance, the power gap between the people is narrow, and subordinates will rely on their superiors limitedly, and they will be more independent between each other.

In a country with a large power distance, children must obey their parents’ discipline, orders between children and adults are emphasized, independent behaviors are discouraged, and it is understood that respecting parents and elders is a virtue. People in the wider society expect others to treat themselves the same way. For people who grow up in an environment with a lower level of power distance, parents and children treat each other equally. There is a relationship of interdependence between those with a higher level of power and those with a lower level of power, and everyone has their value in existence regardless of their status. Therefore, in a country with a lower level of power distance, it is not believed that the level of education will affect the power value of the people.

#### 2.1.2. Individualism/Collectivism

Hofstede [23] used the individualism index score (IDV) to distinguish the relationship; those with higher scores belong to individualistic countries, and the relationship between people is loose. A low scorer relates to a collectivist state, where each member forms a solidarity with other members, protects them for life, and takes care of themselves and their family in exchange for their loyalty.

In a collectivist society, people are born into large families or other groups. From an early age, they must learn to think from a group perspective. In order to maintain good relations with other members, members of the group must continue to protect each other, nurture loyalty with each other, maintain harmony, and avoid conflict. An individualistic society emphasizes personal feelings, and the opinions of others are not important. People learn to think about problems from a personal perspective since childhood. What matters is the individual’s identification with themselves, which is a manifestation of honesty. Conflict is beneficial. Everyone has to take care of themselves and their family when they grow up. Endangering others is seen negatively, and those who do will often lose self-esteem.

#### 2.1.3. Masculinity/Femininity

This is the degree to which gender roles and expectations are differentiated in different countries. The masculine style emphasizes the traditional concept that masculinity should be valued. Hofstede [23] used the masculinity index (MAS) to show that a man may have feminine behavior, and a woman may also have masculine behavior, which means that their behavior differs from general social behavior.

A country with a feminine style teaches the people not to be ambitious and to maintain a humble attitude. Warm interpersonal relationships are very important. The mainstream value of society is to care about others and be conservative; even men should have a gentle disposition, value relationships with each other, and have compassion for the weak. Parents care about facts and emotions, they do not stop boys from crying, as boys do not need to cry because they are afraid of being seen as cowardly by others, and they do not allow people to use violence.

In a country with a masculine style, it is assumed that men are confident, ambitious, and strong; women are assumed to be gentle and caring in their relationships, and everyone has such expectations. The mainstream value of society is material progress and satisfaction, money, and other important items. At home, the father handles things and plays the role of arbiter; the mother handles feelings, listens to others express emotions, and heals emotional wounds. Women are allowed to cry, but men are not allowed; men can fight back against others’ attacks, and women are not allowed to behave violently.

#### 2.1.4. Uncertainty Avoidance

This is the level of anxiety which people within a society feel about the uncertainty of the future. Hofstede [23] used the society’s tolerance for ambiguity as an index to measure the uncertainty avoidance index (UAI). Reducing uncertainty can calm people’s minds.

In countries with strong avoidance of uncertainty, people learn how to avoid dangerous things from childhood. They know that non-standard behaviors can be dangerous, so they will avoid danger and taboos. In the face of unclear and taboo situations, there are strict rules to limit children’s behavior; in the face of an unknown future, there is a strong sense of tension, anxiety, fear of unclear environments and risks, and even aggressive and emotional reactions. In order to avoid being in an environment of uncertainty, rules that are applicable will be formulated one by one. For countries with weak avoidance of uncertainty, uncertainty is a common phenomenon in life. People can stay calm in an uncertain and unclear environment, they will not be nervous about the uncertain environment in their life, and they do not think that uncertainty is dangerous or taboo.

#### 2.1.5. Long-Term/Short-Term Orientation

Confucian ideological dynamism refers to the difference between long-term and short-term tendencies in life, measured by long-term tendencies (LTO). Hofstede and Bond [18] think these values are related to the teachings of Confucius. Long-term-oriented values are future-oriented and relatively dynamic; short-term-oriented values are related to the past and present and are relatively static. Among them: (1) Long-term orientation, which is biased towards the value of Confucian ideological dynamics, involves people facing the future, believing that the traditions of the past will change with the times, and observing things from a dynamic point of view, so there will be room for everything. (2) The value of short-term orientation lies in people paying more attention to current interests and pleasures, and hoping to see results in a short time. Quick success is more urgent and cannot be delayed.

#### 2.1.6. Indulgence/Restraint (IVR)

The extent to which members of society accept their basic needs and desires to enjoy life. Indulgence represents the basic normal desire to enjoy the pleasures of life, allowing unrestrained satisfaction, which is an unrestricted society; constraints reflect the need to control the enjoyment of life and manage with strict social norms, which is a restricted society [24].

Countries that indulge cultural characteristics have more people who feel very happy, attach importance to friends and leisure, have high life autonomy, are outgoing and optimistic, have a positive attitude (not cynical), and feel good about themselves. People are highly receptive to foreign music and movies, are satisfied with family life, housework is shared by both parents, they actively participate in physical exercise, and use email and the Internet to interact with others. Constrained societies are the opposite; social groups will have greater constraints on themselves [24].

In sum, Hofstede’s national cultural theory measures the national cultural preferences of a specific country from six different dimensions, and provides a benchmark for people to identify and understand cultural phenomena. It is an effective tool for comparing and analyzing different cultures, and can help managers of international companies to quickly grasp consumer business opportunities and provide a foundation for cross-cultural management.

### 2.2. National Culture and COVID-19

According to the characteristics of different cultures in different countries, it is difficult for international companies who encountered the COVID-19 pandemic to formulate effective strategies to avoid the risks caused by uncertainty [13]. Therefore, the uncertainty brought about by the COVID-19 pandemic will affect the globalization path of enterprises, the choice of entry mode, and the speed of international expansion. At this time, only minimizing risks can improve the operating performance for enterprises [25]. The COVID-19 pandemic has had a great impact on the development of e-commerce in mainland China, and consumers are willing to buy food online [7].

Sohaib et al. [26] believe that culture can affect consumers’ online shopping behavior. They analyzed the online shopping behavior of Australian consumers based on Hofstede’s [18] uncertainty avoidance. The results show that uncertainty avoidance does affect consumers’ online shopping behavior [27]. Urbaczewski and Lee [16] tracked the confirmed cases of COVID-19 among people in mainland China, Germany, Italy, Singapore, South Korea, and the United States. They found that national culture has a high and significant correlation with the reduction of COVID-19 [16].

At present, although scholars have analyzed why the COVID-19 pandemic is better controlled in some countries than others based on the individualism/collectivism in Hofstede’s national cultural dimensions [4,10], they believe that different countries will have different responses to the COVID-19 pandemic. For countries that have collectivist cultural values, their people will sacrifice individual freedom for the interests of the group. Thus, as large-scale social coordination was a key coping mechanism during the pandemic, it is possibly related to more collectivist cultures in mainland China, South Korea, Taiwan, and Singapore. This is because, when people in a collectivist society are at risk, they will take more actions to protect individuals or communities [5], and will also strengthen the importance of solidarity through the media as a buffer to prevent the spread of the COVID-19 pandemic. The barrier to transmission helps fight the outbreak, so collectivist societies may be one of the reasons why these countries are doing so well [5]. It is also why collectivism in mainland China is better at blocking viruses [28].

In contrast, an individualistic society places more emphasis on individuals and freedoms and considers group interests less [5], thus reducing the effectiveness of large-scale social coordination, and social distancing measures were relatively ignored during the COVID-19 pandemic. Therefore, cultural differences are important, and managers need to understand how cultural differences may affect the way people process information and make decisions [5].

However, the above studies related to the COVID-19 pandemic and culture mostly focus on the dimensions of individualism/collectivism and uncertainty avoidance of Hofstede’s national cultural theory perspective. Meanwhile, Shetty et al. [14] has studied an overview of five of Hofstede’s dimensions (PDI, IDV, MAS, UAI, and LTO) and their impact on the implementation of COVID-19 control strategies. Their study used a case analysis of four countries: India, the United Kingdom, the United States of America, and Poland, to illustrate the interplay between culture and COVID-19 control strategies, and they demonstrated that cultural differences can significantly impact the success of COVID-19 control strategies [14]. In addition, Timo et al. [15] based their study on Hofstede’s cultural dimensions in order to study its correlation with COVID-19 data. They found the pandemic reached different countries at different times, and the role of socio-economic and cultural factors can be drawn only after the pandemic [15]. Many scholars also adopted Hofstede’s cultural dimensions [3,6,8,9,11,12,17] to study the correlation between Hofstede’s cultural dimensions and COVID-19 data. However, these studies did not focus on the difference of Hofstede’s cultural dimensions to explore the gap between different WHO regions.

For this reason, we believe that the analysis level should be expanded to six dimensions of Hofstede’s national cultural theory to further understand why the severity in the monitoring and management mechanisms of the COVID-19 pandemic in different countries of the WHO regions have a large gap. Therefore, the results of this study should be different with previous studies. For this reason, this study can improve the ability of pandemic control and serve as a basis and reference for future academics, practice, medicine, and government.

## 3. Meta-Research Method

This study uses the narrative approach of the meta-research method [29,30] based on the WHO regions of Africa (AFRO), Europe (EURO), the Americas (AMRO), the Western Pacific (WPRO), South East Asia (SEARO), and the Eastern Mediterranean (EMRO) to integrate and analyze COVID-19 data. Therefore, in this study we collected various data on COVID-19 in countries around the world before vaccines were administered from 22 February 2020 to 20 February 2021, including: (1) Cumulative_cases (CC). (2) Cases—cumulative total per 1 million population (CC-PM). (3) Cases—newly reported in last 7 days (NC-7A). (4) New_cases (NC). (5) Cumulative_deaths (CD). (6) Deaths—cumulative total per 1 million population (CD-PM). (7) Deaths—newly reported in last 7 days (ND-7A). (8) New_deaths (ND) (see Table 1). Then, based on the time axis and the six dimensions of Hofstede’s national culture, the difference between different WHO regions were compared.

### 3.1. Data Collection

The expectation of this study was to divide the world into different national cultures, and to understand the impact and differences of COVID-19 in different WHO regions. In order to ensure of accuracy of the research results through the data collected in this study, it was necessary to collect relevant data and literature in a comprehensive manner. The data sources for this study include: the World Health Organization [31], the Global Change Data Lab (GCDL) [32], the Nation Center for High-Performance Computing [33], and the CSSEGISandData [34].

The scope of this research included 140 countries affected by COVID-19, according to the statistics of the WHO; however, because the data of Hofstede’s six dimensions of national culture only include 117 countries, this research uses Hofstede’s national culture of 117 countries as the basis for the analysis.

### 3.2. Data Analysis

First of all, due to different spread speeds of COVID-19 in various countries around the world in the early stage, and in order to understand the differences in the impact of time on the epidemic situation in different regions of the WHO, this study mostly uses the month as the unit of analysis for the data collected from 22 February 2020, and contains the data from the following time points: 22 February 2020, 22 March 2020, 22 April 2020, 22 May 2020, 22 June 2020, 22 July 2020, 22 August 2020, 22 September 2020, 22 October 2020, 22 November 2020, 11 December 2020, 31 December 2020, 11 January 2021, 21 January 2021, 30 January 2021, 13 February 2021, and 20 February 2021.

Secondly, we take the six dimensions of Hofstede’s national culture as the units of analysis, including: power distance (PDI), individualism/collectivism (IDV), masculinity/femininity (MAS), uncertainty avoidance (UAI), long-term/short-term orientation (LTO), and indulgence/restraint (IVR).

In this study, we take the time axis and the six dimensions of Hofstede’s national culture as the benchmark for analysis, and conduct correlation analyses with following eight data items from each WHO region: Cumulative_cases, Cases—cumulative total per 1 million population, Cases—newly reported in last 7 days, New_cases (Cases—newly reported in last 24 h), Cumulative_deaths, Deaths—cumulative total per 1 million population, Deaths—newly reported in last 7 days, and New_deaths (Deaths—newly reported in last 24 h). Analysis items: (1) Analysis of significant differences in the correlation of eight data from the same WHO region at different times in each of Hofstede’s national cultural dimensions. (2) Analysis of significant difference in correlation of eight data from different WHO regions at the same time in each of Hofstede’s national cultural dimensions.

## 4. Results and Discussion

### 4.1. The Mean Value of Hofstede’s Six Dimensions in Each WHO Region (See Figure A1)

The mean value of Hofstede’s six dimensions (PDI, IDV, MAS, UAI, LTO, and IVR) in each WHO region is based on the score of each of Hofstede’s dimensions in each country. Meanwhile, some countries do not have LTO and IVR scores; thus, the LTO and IVR score of these countries has to be ignored in this study when the mean value of the LTO and IVR in each WHO region is calculated.

(1) PDI mean value: The PDI mean value of EMRO is 77.93, the PDI mean value of SEARO is 76.86, the PDI mean value of AFRO is 71.69, the PDI mean value of WPRO is 66.33, the PDI mean value of AMRO is 66.13, and the PDI mean value of EURO is 60.00. The higher the scores of the PDI, the larger the power distance of the countries [18].

All of the EMRO countries have higher PDI scores. Most of the AFRO countries have higher PDI scores, except South Africa. All of the SEARO countries and most of the WPRO countries (except Australia and New Zealand) have higher PDI scores. Most of the AMRO countries have higher PDI scores, except Argentina, Canada, Costa Rica, Jamaica, Trinidad and Tobago, and the USA; thus, this region also has a high PDI mean value too. The PDI mean value of the EURO region is also high because only 17 countries’ PDI scores are lower than 50, and the other 22 countries’ PDI scores are higher than 60.

(2) IDV mean value: The IDV mean value of EURO is 49.00, the IDV mean value of WPRO is 33.92, the IDV mean value of EMRO is 31.36, the IDV mean value of SEARO is 31.29, the IDV mean value of AMRO is 28.04, and the IDV mean value of AFRO is 27.50. The higher the IDV scores, the stronger the individualism of the countries [18].

Most of the EURO countries have higher IDV scores, except Albania, Armenia, Azerbaijan, Belarus, Bosnia and Herzegovina, Bulgaria, Croatia, Georgia, Greece, Kazakhstan, Moldova, Montenegro, North Macedonia, Portugal, Romania, Russia, Serbia, Slovenia, Turkey, and Ukraine. Most of the WPRO countries have lower IDV scores, except Australia and New Zealand. All of the EMRO and SEARO countries have lower IDV scores. Most of the AMRO countries have lower IDV scores, except Canada and the USA. Most of the AFRO countries have lower IDV scores, except South Africa.

(3) MAS mean value: The MAS mean value of WPRO is 55.75, the MAS mean value of EMRO is 51.43, the MAS mean value of AMRO is 49.74, the MAS mean value of EURO is 44.84, the MAS mean value of AFRO is 43.75, and the MAS mean value of SEARO is 39.00. The higher the MAS scores, the higher the masculinity of the countries [18].

Most of the WPRO countries have higher MAS scores, except Fiji, South Korea, Singapore, Taiwan, and Vietnam. Most of the EMRO countries have higher MAS scores, except Egypt, Iran, Jordan, Kuwait, and Tunisia. Half of the AMRO countries have lower MAS scores, except Argentina, Canada, Colombia, the Dominican Republic, Ecuador, Jamaica, Mexico, Puerto Rico, Trinidad and Tobago, the United States, and Venezuela. Most of the EURO countries have lower MAS scores, except Albania, Armenia, Austria, Azerbaijan, Belgium, Czech Republic, Georgia, Germany, Greece, Hungary, Ireland, Italy, Kazakhstan, Luxembourg, Poland, Slovakia, Switzerland, and the United Kingdom. Most of the AFRO countries have lower MAS scores, except Burkina Faso, Ethiopia, Kenya, Nigeria, and South Africa. Most of the SEARO countries have lower MAS scores, except Bangladesh and India.

(4) UAI mean value: The UAI mean value of EURO is 75.22, the UAI mean value of AMRO is 72.22, the UAI mean value of EMRO is 71.43, the UAI mean value of AFRO is 54.26, the UAI mean value of WPRO is 47.58, and the UAI mean value of SEARO is 46.43. The higher the UAI scores, the higher the uncertainly avoidance of the countries [18].

Most of the EURO and AMRO regions’ countries have higher UAI scores (except Denmark, Ireland, Sweden, the United Kingdom in EURO, and Canada, the Dominica Republic, Jamaica, Puerto Rico, and the United States in AMRO) than the WPRO and SEARO regions (except Australia, Japan, South Korea, and Taiwan in WPRO, and Bangladesh and Thailand in SEARO). At the same time, all of the EMRO countries have higher UAI scores, and the UAI scores of the AFRO countries are higher than 50 (except Mozambique, Namibia, and South Africa).

(5) LTO mean value: The LTO mean value of WPRO is 61.82, the LTO mean value of EURO is 57.71, the LTO mean value of SEARO is 47.40, the LTO mean value of AFRO is 23.83, the LTO mean value of EMRO is 22.90, and the LTO mean value of AMRO is 22.00. The higher the LTO scores, the higher the long-term orientation of the countries [18].

Most of the WPRO and EURO countries have higher LTO scores (except Australia, Malaysia, New Zealand, and the Philippines in WPRO, and Belgium, Denmark, Finland, Georgia, Greece, Iceland, Ireland, Israel, Malta, Norway, Poland, Portugal, Slovenia, Spain, and Turkey in EURO). In the SEARO region, India and Indonesia were the two countries with higher LTO scores; all of the AFRO, EMRO, and AMRO regions’ countries have lower LTO mean values.

(6) IVR mean value: The IVR mean value of AMRO is 69.94, the IVR mean value of AFRO is 54.45, the IVR mean value of WPRO is 44.27, the IVR mean value of EURO is 39.11, the IVR mean value of SEARO is 32.25, and the IVR mean value of EMRO is 26.67. The higher the IVR scores, the higher the indulgence of the countries [20].

Most of the AMRO countries have higher IVR scores, except Bolivia and Peru. The IVR mean value of the AFRO region is over 50; however, the IVR scores of six countries (Algeria, Burkina Faso, Ethiopia, São Tomé and Príncipe, Tanzania, and Zambia) are lower than 50. Most of the WPRO countries have lower IVR scores, except Australia and New Zealand. Most of the EURO countries have lower IVR scores, except Austria, Belgium, Denmark, Finland, Greece, Iceland, Ireland, Luxembourg, Malta, the Netherlands, Norway, Sweden, Switzerland, and the United Kingdom. All of the SEARO and EMRO countries have lower IVR scores.

### 4.2. Hofstede’s Six Dimensions and COVID-19 Data

#### 4.2.1. The Power Distance (PDI) Correlation with COVID-19 Data (Figure A2, Figure A8, Figure A11 and Figure A12)

During the period from 22 February 2020 to 20 February 2021, PDI only had negative significant correlation with the CC and ND-7A on 22 April 2020; PDI had negative significant correlation with NC-7A and NC on 22 March 2020. Meanwhile, PDI had negative significant correlation with CD on 22 April 2020 and 22 May 2020; PDI had negative significant correlation with CD-PM on 22 April 2020, 22 May 2020, and 22 June 2020; PDI had negative significant correlation with ND on 22 April 2020 and 31 December 2020; and PDI had negative significant correlation with CD-PM on 22 April 2020, 22 May 2020, and 22 June 2020. Finally, PDI had negative significant with CC-PM on 22 March 2020, 21 January 2021, 30 January 2021, 13 February 2021, and 20 February 2021.

These results show that the lower PDI scores could be an important factor in the increasing global human infection and even death from COVID-19 in the early stage of the COVID-19 pandemic. The lower PDI scores could be a critically significant factor leading to an increase in the CC-PM even in the first year of the COVID-19 pandemic. For this reason, the differences in each WHO region due to PDI scores will be discussed in the next section of this study.

##### PDI and the WHO Europe (EURO) Region

PDI only had negative significant correlation with CC-PM on 22 February 2020, 22 March 2020, and 22 April 2020. Meanwhile, PDI had positive significant correlation with ND-7A on 22 July 2020 and 22 August 2020 and PDI had positive significant correlation with ND on 22 August 2020.

The results show that the lower PDI score is a critical factor in increasing the CC-PM, and a decreased ND-7A and ND of EURO people in the early stage of the COVID-19 pandemic. It is probably because, although EURO has the lowest PDI mean value, many EURO countries have relatively higher PDI scores, and many people from these countries are Christians or Catholics. Thus, even the governments recognized the severe and widespread situation of COVID-19, but people were not willing to follow the isolation policies of their governments. In contrast, the number of ND-7A and ND will decrease; it is probably because EURO has relatively sound medical systems.

##### PDI and the WHO Africa (AFRO) Region

PDI had a negative significant correlation with the CC, CD, and ND-7A from 22 June 2020 to 20 February 2021, except the CD on 22 September 2020. PDI had a negative significant correlation with CC-PM and CD-PM from 22 July 2020 to 20 February 2021, except the CD-PM on 22 September 2020. Meanwhile, PDI had a negative significant correlation with NC-7A, NC, and ND from 22 May 2020 to 20 February 2021, except the NC-7A on 20 February 2021, the NC on 22 September 2020 and 13 February 2021, and the ND on 22 October 2020 and 22 November 2020.

The AFRO region was the last place to be infected with COVID-19 and has fewer data in the first three months (22 February 2020, 22 March 2020, and 22 April 2020), and the PDI scores do not have a significant correlation with COVID-19 data. However, when the number of people infected with COVID-19 increased in the middle and end of the first year of the COVID-19 pandemic period, the higher PDI scores indeed have significantly decreased COVID-19 data from people. Therefore, the results can probably be explained because most of the AFRO countries have higher PDI scores (except South Africa, which was colonized by the Netherlands and the United Kingdom; therefore, its PDI score is 49).

##### PDI and the WHO Americas (AMRO) Region

The mean number of AMRO people infected with COVID-19 in the first year of the COVID-19 pandemic period is the highest in the world (see Figure A11). However, PDI only had a negative significant correlation with CC and NC-7A on 22 February 2020, and PDI had a negative significant correlation with CD-PM on 22 March 2020.

Most of the AMRO countries have relatively higher PDI scores; however, these countries are located in the South America. Some North American, Central American, and Caribbean countries have lower PDI scores; thus, the AMRO PDI mean value (66.13) also is reduced. The USA (United States of America) has a very low PDI score, and the USA had the highest number of people infected with COVID-19 in the first year of the COVID-19 pandemic since 22 March 2020; but, lower PDI scores in the region did increase alongside the number of people infected with COVID-19 in the early stages of the COVID-19 pandemic (22 February 2020), but has no significant correlation with COVID-19 data since 22 March 2020 in the first year of COVID-19 pandemic period. This is probably because most countries in the AMRO region have relatively higher PDI scores, and so PDI has no significant correlation with COVID-19 data in this region.

##### PDI and the WHO Eastern Mediterranean (EMRO) Region

PDI had a negative significant correlation with the ND-7A and ND on 22 May 2020. PDI had a negative significant correlation with the CD on 22 July 2020. PDI had a negative significant correlation with the CD-PM on 22 September 2020 and 22 October 2020. PDI had a negative significant correlation with the NC-7A and NC on 22 November 2020, 11 December 2020, and 31 December 2020. PDI had a negative significant correlation with the ND-7A and ND from 22 November 2020 to 20 February 2021, except the ND on 22 November 2020.

People in this region had been infected with COVID-19 since the beginning of the COVID-19 pandemic period. All of the EMRO countries are Muslim countries, and the EMRO has the highest PDI mean value (77.93). The result is probably because people have to respect their governments’ isolation policies for suppressing the spread of COVID-19; thus, the PDI indeed will decrease the COVID-19 spread and death number of this region at the end of the first year of the COVID-19 pandemic period.

In addition, many people were infected with COVID-19 and died in the end of the first year of the COVID-19 pandemic period. This is probably because there were local wars in many EMRO countries, and it was difficult to receive timely treatment for COVID-19.

##### PDI and the WHO Western Pacific (WPRO) and South East Asia (SEARO) Regions

PDI had no significant correlation with all of the CC, CC-PM, NC-7A, NC, CD, CD-PM, ND-7A, and ND through the first year of the COVID-19 pandemic period. Therefore, the PDI is not an important factor in influencing in the first year of COVID-19 in these two regions.

On the one hand, COVID-19 was first detected in mainland China (PRC); therefore, in just the first month, the mean of the CC and CD in the WPRO is the highest number of people infected with COVID-19 in the world.

On the other hand, due to the isolation policy of governments and the fact that most people from this region are Buddhist, and this region has a relatively higher PDI mean value (66.33). (Two countries in the anglosphere, Australia and New Zealand, are classified by the WHO in the WPRO region, and the PDI values of these two countries (38 and 22) are very low; thus, the PDI mean value of the WPRO is reduced by these two countries.) This shows that people will respect the order of their government’s isolation policy. Therefore, very few WPRO people were infected with COVID-19 in the first year of the COVID-19 pandemic period; then, the PDI has no significant correlation with COVID-19 data in this region.

Many people were infected with COVID-19 in the end of the first year of the COVID-19 pandemic period in the SEARO region, most of people in this region are Buddhist, and this region has a very high PDI mean value (76.86); however, the PDI has no significant correlation with COVID-19 data in this region. Thus, the result is probably because the medical systems of many countries in this region are weak.

#### 4.2.2. The Individualism/Collectivism (IDV) Correlation with COVID-19 Data (Figure A3, Figure A8, Figure A11 and Figure A13)

During the period from 22 February 2020 to 20 February 2021, IDV had a positive significant correlation with the CC from 22 March 2020 in the first year of the COVID-19 pandemic period. IDV had a positive significant correlation with the CC-PM on 22 March 2020, 22 May 2020, and from 22 November 2020 to 20 February 2021. IDV had a positive significant correlation with the CD from 22 April 2020 in the first year of the COVID-19 pandemic period, except for 22 September 2020. IDV had a positive significant correlation with the NC-7A and NC from 22 March 2020 in the first year of the COVID-19 pandemic period, except for the NC-7A on 22 June 2020, 22 August 2020, and 22 September 2020. Meanwhile, IDV had a positive significant correlation with the CD-PM from 22 March 2020 in the first year of the COVID-19 pandemic period, except for 22 August 2020, 22 September 2020, 22 October 2020, and 22 November 2020. Finally, IDV had a positive significant correlation with the ND-7A and ND from 22 March 2020 in the first year of the COVID-19 pandemic period, except for 22 June 2020, 22 July 2020, 22 August 2020, and 22 September 2020.

These results show that the higher IDV score could be a critical factor in the increase in the global number of people infected with COVID-19. For this reason, the differences in each WHO region due to IDV scores will be discussed in this study.

##### IDV and the WHO Europe (EURO) Region

IDV had a positive significant correlation with the CC on 22 February 2020, 22 March 2020, 22 April 2020, 11 January 2021, and 21 January 2021. IDV had a positive significant correlation with the CC-PM on 22 March 2020, 22 February 2020, 22 March 2020, and 22 May 2020. IDV had a positive significant correlation with the NC-7A and NC on 22 March 2020, 22 October 2020, 31 December 2020, 11 January 2021, 21 January 2021, 13 February 2021, and 20 February 2021. Additionally, IDV had a positive significant correlation with the CD from 22 April 2020 in the first year of the COVID-19 pandemic period. IDV had a positive significant correlation with the CD-PM from 22 March 2020 to 22 August 2020. Finally, IDV had a positive significant correlation with the ND-7A on 22 April 2020 and 22 May 2020, and from 11 January 2021 to 20 February 2021. IDV had a positive significant correlation with the ND on 22 April 2020, 22 May 2020, 31 December 2020, 11 January 2021, 21 January 2021, 13 February 2021, and 20 February 2021.

The results show that a higher IDV score is a critical factor in increasing the CD in the first year of the COVID-19 pandemic period, and increasing the NC-7A, NC, ND-7A, and ND at the end of the first year of the COVID-19 pandemic period. It is probably because many EURO countries have very high IDV scores, especially in West Europe, and Individualism is the dominate assertion of people from the EURO region. Thus, they cannot be restricted through wearing a mask, social distance policies, or isolation policies of their governments. For this reason, the mean number of EURO people with COVID-19 increased throughout the first year of the COVID-19 pandemic period.

##### IDV and the WHO Africa (AFRO) Region

IDV had a positive significant correlation with the CC, NC, and NC-7A from 22 March 2020 in the first year of the COVID-19 pandemic period, except the NC on 22 September 2020. IVD had a positive significant correlation with the CC-PM on 22 March 2020 and from 22 May 2020 to 20 February 2021. Meanwhile, IDV had a positive significant correlation with CD, ND-7A, and ND from 22 May 2020 to 20 February 2021, except the ND-7A on 22 September 2020 and the ND on 22 October 2020. Finally, IDV had a positive significant correlation with the CD-PM on 22 June 2020, 22 July 2020, and 22 August 2020, and from 22 October 2020 to 20 February 2021.

The mean number of AFRO people infected with COVID-19 is relatively lower than other WHO regions, probably because the governments of the AFRO region cannot provide valid Polymerase chain reactions (PCRs) and quick tests. Even though most people from the AFRO region emphasize collectivistic culture, this region has the lowest IDV score, and the number of people infected with COVID-19 will decrease. In light of this reason, IDV was a significant factor throughout the first year of the COVID-19 pandemic period.

##### IDV and the WHO Americas (AMRO) Region

IDV had a positive significant correlation with the CC, NC-7A, and NC from 22 February 2020 in the first year of the COVID-19 pandemic period. IDV had a positive significant correlation with the CC-PM on 22 February 2020, 22 March 2020, 22 April 2020, 22 May 2020, 21 January 2021, and 13 February 2021. Meanwhile, IDV had a positive significant correlation with the CD from 22 February 2020 in the first year of the COVID-19 pandemic period. IDV had a positive significant correlation with the ND-7A and ND on 22 March 2020, 22 April 2020, and 22 May 2020, the ND-7A from 22 September 2020 to 22 February 2020, and the ND from 22 October 2020 to 22 February 2020. Finally, IDV had a positive significant correlation with the CD-PM on 22 March 2020 and 22 April 2020.

Most of the AMRO countries had a large number of people infected with COVID-19 in the first year of the COVID-19 pandemic period, and this region’s IDV score range of each country is 6~91. Therefore, a higher IDV score is indeed a very critical factor in increasing the number of people infected with COVID-19 in the AMRO region through the first year of the COVID-19 pandemic period.

##### IDV and the WHO Eastern Mediterranean (EMRO) Region

IDV only had a negative significant correlation with the NC-7A and NC on 22 June 2020. IDV had a positive significant correlation with the CD-PM at the end of the first year of the COVID-19 pandemic period (from 11 January 2021 to 20 February 2021).

People in this region were infected with COVID-19 from the beginning of the COVID-19 pandemic period. All of the EMRO countries are Muslim countries, and the EMRO has a relatively low IDV mean value (31.36). The lower IDV score increases the NC-7A and NC on 22 June 2020 probably because some countries have many people from Western societies, but the phenomenon was decreased at the end of the first year of the COVID-19 pandemic period. Finally, in the EMRO region people do not emphasize individualism; then, the CD-PM has indeed decreased at the end of the first year of the COVID-19 pandemic period.

##### IDV and the WHO Western Pacific (WPRO) and South East Asia (SEARO) Regions

IDV had no significant correlation with any of the CC, CC-PM, NC-7A, NC, CD, CD-PM, ND-7A, and ND through the first year of the COVID-19 pandemic period. Therefore, the IDV will not be an important factor in influencing the first year of the COVID-19 pandemic period in these two regions.

Because of the isolation policies of governments and most people of these two regions are Buddhist, this region has a relatively low IDV mean value (WPRO: 33.92, SEARO: 31.29); thus, the people with collectivistic cognition have sacrificed their own interests to protecting their family, relatives, friends, and society. Therefore, far fewer people were infected with COVID-19 in this region in the first year of the COVID-19 pandemic; therefore, the PDI has no significant correlation with COVID-19 data in this region. (Two countries in the anglosphere, Australia and New Zealand, are classified by the WHO in the WPRO region, and the IDV values of these two countries (90 and 79) are very high; thus, the IDV mean value of WPRO is increased by these two countries.)

#### 4.2.3. The Masculinity/Femininity (MAS) Correlation with COVID-19 Data (Figure A4, Figure A9, Figure A11 and Figure A14)

During the period from 22 February 2020 to 20 February 2021, MAS had a positive significant correlation with the ND-7A on 22 November 2020, 31 December 2020, 11 January 2021, 21 January 2021, and 30 January 2021. MAS had a positive significant correlation with the ND on 22 November 2020, 31 December 2020, and 21 January 2021.

These results show that the higher MAS scores increase global COVID-19 deaths only at the end of the first year of the COVID-19 pandemic period; therefore, MAS is probably not a critical factor in influencing the global number of people to be infected with COVID-19 in the first year of the COVID-19 pandemic period. The differences in each WHO region due to MAS scores will be discussed in the next sections of this study.

##### MAS and the WHO Africa (AFRO) Region

MAS had a positive significant correlation with the NC-7A on 22 August 2020, 22 September 2020, 22 October 2020, 22 November 2020, 13 February 2021, and 20 February 2021. Meanwhile, MAS had a positive significant correlation with the NC on 22 August 2020, 22 September 2020, 22 October 2020, 22 November 2020, 11 December 2020, and 20 February 2021. Finally, MAS had a positive significant correlation with the ND on 22 August 2020, 22 September 2020, 22 October 2020, and 22 November 2020.

A high MAS score means that the society emphasizes the traditional concept that masculinity should be valued, and most countries in AFRO have relatively lower MAS scores of the six WHO regions. Therefore, only few dates of higher MAS scores have increased the NC-7A, NC, and ND in the middle of the first year of the COVID-19 pandemic period, and MAS only increases the NC-7A and NC at the end of the first year of the COVID-19 pandemic period in the AFRO region. Thus, MAS is not an important factor in influencing the number of AFRO people infected with COVID-19 in the first year of the COVID-19 pandemic period.

##### MAS and the WHO Western Pacific (WPRO) Region

MAS had a positive significant correlation with the NC-7A on 22 November 2020, 11 December 2020, 31 December 2020, 11 January 2021, and 21 January 2021. MAS had a positive significant correlation with the NC on 22 November 2020, 11 January 2021, and 21 January 2021. Meanwhile, MAS only had a positive significant correlation with the CD on 20 February 2021. Finally, MAS had a positive significant correlation with the ND-7A on 22 April 2020 and 22 May 2020, and from 22 November 2020 to 20 February 2021. MAS had a positive significant correlation with the ND on 22 April 2020, 22 May 2020, and 11 December 2020, and from 11 January 2021 to 13 February 2021.

The higher MAS score increases the NC-7A, NC, ND-7A, and ND at the end of the first year of the COVID-19 pandemic period, increases the ND-7A and ND in the beginning of the first year of the COVID-19 pandemic period, and increases the CD in the last month of the first year of the COVID-19 pandemic period. The MAS mean value is 55.75 in the WPRO region, which has the highest value; thus, many countries highly emphasized the traditional concept that masculinity should be valued in the WPRO region. For this reason, this region has the most influential COVID-19 data by MAS of the six WHO regions. Therefore, the MAS score is probably a factor in increasing COVID-19 data of this region, but not a critical factor in the first year of the COVID-19 pandemic period.

##### MAS and the WHO South East Asia (SEARO) Region

MAS had a positive significant correlation with the CCPM on 22 July 2020 and 22 August 2020. Meanwhile, MAS had a positive significant correlation with the CD-PM from 22 June 2020 to 22 October 2020. Finally, MAS only had a positive significant correlation with the ND-7A and ND on 22 May 2020.

The MAS mean value is 39.00 for the SEARO region, which was the lowest value; thus, only a few countries emphasized the traditional concept that masculinity should be valued in the SEARO region. Therefore, the MAS score only increases the CD-PM in the first five months, the CC in the middle two months, and the ND-7A and ND on 22 May 2020. Therefore, MAS is also not a critical factor in increasing COVID-19 data in this region in the first year of the COVID-19 pandemic period.

##### MAS and the WHO Eastern Mediterranean (EMRO) Region

The MAS mean value (51.43) in EMRO is second highest of the six WHO regions, and these countries highly emphasized the traditional concept that masculinity is valued too. However, MAS only had a negative significant correlation with the NC on 22 August 2020, and MAS is indeed not a significant factor in increasing the number of EMRO people infected with COVID-19.

##### MAS and the WHO Europe (EURO) and Americas (AMRO) Regions

MAS had no significant correlation with any of the CC, CC-PM, NC-7A, NC, CD, CD-PM, ND-7A, and ND through the first year of the COVID-19 pandemic period. Therefore, MAS is not an important factor in influencing the first year of COVID-19 data in these two regions.

The results show that the MAS score is not a critical factor in influencing all of the COVID-19 data in the first year of the COVID-19 pandemic period in these two regions.

#### 4.2.4. The Uncertainty Avoidance (UAI) Correlation with COVID-19 Data (Figure A5, Figure A9, Figure A11 and Figure A15)

During the period from 22 February 2020 to 20 February 2021, UAI only had a negative significant correlation with the CC-PM on 22 February 2020, but UAI had a positive significant correlation with the CC-PM from 22 July 2020 to 20 February 2021. Meanwhile, UAI had a positive significant correlation with the ND-PM from 22 August 2020 to 20 February 2021.

UAI means the society’s tolerance for ambiguity, and a higher UAI society can tolerate more ambiguity. These results show that a lower UAI score increases the global number of people infected with COVID-19 at the first month of the COVID-19 pandemic period. However, since 22 July 2020 (a few months after 22 February 2020), a higher UAI score significantly increased the correlation with the number of people infected with COVID-19 and death in the first year of the COVID-19 pandemic period. It is necessary to explore why UAI trends in the opposite direction to the global number of people infected with COVID-19, and the differences in each WHO region due to UAI scores will be discussed in the next section of this study.

##### UAI and the WHO Africa (AFRO) Region

UAI only had a positive significant correlation with the CC-PM and CD-PM on 22 April 2020 and 22 May 2020. Meanwhile, UAI only had a positive significant correlation with the CD on 22 March 2020 and 22 April 2020. UAI only had a positive significant correlation with the ND-7A on 22 March 2020. UAI only had a positive significant correlation with the NC on 22 April 2020.

The UAI mean value (54.56) of AFRO is relatively low for the six WHO regions. This means that AFRO societies are not societies which tolerate ambiguity. Meanwhile, the relatively low UAI score has only increased some COVID-19 data in the beginning three months of the first year of the COVID-19 pandemic period. Therefore, UAI is not an important factor in influencing the number of AFRO people infected with COVID-19 in the first year of the COVID-19 pandemic period.

##### UAI and the WHO Western Pacific (WPRO) Region

UAI only had a positive significant correlation with the NC-7A on 31 December 2020. Meanwhile, UAI only had a positive significant correlation with the ND-7A on 31 December 2020 and 11 January 2021. UAI only had a positive significant correlation with the ND on 22 April 2020, 11 December 2020, and 31 December 2020.

All of the global countries did not have enough knowledge for people to avoid being infected with COVID-19 at that time. The WPRO region has a relatively low UAI mean value (47.58) for the six WHO regions, and this region cannot tolerate ambiguity. However, because COVID-19 is a horrible disease, the governments in this region provided lockdown strategies to reduce the spread of COVID-19. Meanwhile, the lower UAI score of this region only decreases some of the COVID-19 data in the first year, and the results show that the UAI score is probably not an important factor in increasing the number of people infected with COVID-19 in this region in the first year of the COVID-19 pandemic period.

##### UAI and the WHO South East Asia (SEARO) Region

UAI only had a positive significant correlation with the CC-PM on 22 April 2020. Meanwhile, the UAI mean value (46.43) of the SEARO region is the lowest of the six WHO regions, but a lower UAI only slightly decreases the number of people in SEARO infected with COVID-19 in the first year of the COVID-19 pandemic period; thus, UAI is not an important factor for COVID-19.

##### UAI and the WHO Europe (EURO), Americas (AMRO), and Eastern Mediterranean (EMRO) Regions

UAI only had a positive significant correlation with the ND-7A and ND on 22 August 2020 in the EURO region, and the UAI score had no significant correlation with any of the CC, CC-PM, NC-7A, NC, CD, CD-PM, ND-7A, and ND throughout the first year of the COVID-19 pandemic period in both the AMRO and EMRO regions.

The societies of EURO, AMRO, and EMRO could tolerate the ambiguity of COVID-19 in the first COVID-19 pandemic period, because these three regions have the highest UAI mean values (EURO: 75.22; AMRO: 72.22; EMRO: 71.43) in the six WHO regions. However, the higher UAI only increases some of the COVID-19 data in the EURO region, and does not influence the COVID-19 data in the AMRO and EMRO regions. In light of this, the UAI can be ignored as a factor in influencing the first year of the COVID-19 pandemic period in these three WHO regions.

#### 4.2.5. The Long-Term/Short-Term Orientation (LTO) Correlation with COVID-19 Data (Figure A6, Figure A10, Figure A11 and Figure A16)

During the period from 22 February 2020 to 20 February 2021, LTO had a positive significant correlation with the CC and NC on 22 March 2020. Meanwhile, LTO had a positive significant correlation with the CC on 22 February 2020, and UAI had a positive significant correlation with the CC-PM from 11 December 2020 to 20 February 2021. LTO is based on the Confucian ideological dynamism. When people look to the future, they believe that the traditions of the past will change with the times, and observe things from a dynamic point of view, so there will be room for everything. These results show that a higher LTO score significantly increases, but does not decrease, the global number of people infected with COVID-19 in the last three months of the first year of the COVID-19 pandemic period, and only slightly increases the global number of people infected with COVID-19 in the first two months. The differences in each WHO region due to LTO scores will be discussed in the next section of this study.

##### LTO and the WHO Europe (EURO) Regions

LTO only has a negative significant correlation with the CC-PM on 22 May 2020. The EURO region has a relative higher LTO mean value (57.71) because of many eastern EURO countries who have different cultures than western EURO. Thus, they have higher LTO scores than western EURO, and they look to the future more, believing that the traditions of the past will change with the times, and observing things from a dynamic point of view. Meanwhile, the results show that the higher LTO only decreases the number of EURO people infected with COVID-19 in the third month of the COVID-19 pandemic period. However, the LTO is not an important factor in influencing the number of EURO people infected with COVID-19 in the first year of the COVID-19 pandemic period.

##### LTO and the WHO Americas (AMRO) Region

LTO only had a positive significant correlation with NC-7A on 22 June 2020. LTO had a positive significant correlation with the NC on 22 May 2020 and 22 June 2020. Meanwhile, LTO had a positive significant correlation with the ND-7A on 22 June 2020, 22 July 2020, and 22 August 2020. LTO had a positive significant correlation with the ND on 22 June 2020 and 22 August 2020.

The AMRO region has the lowest LTO mean value (22.09) in the six WHO regions, and AMRO people pay more attention to current interests and pleasures; they hope to see results in a short time, and quick success is more urgent and cannot be delayed. However, the higher LTO score only slightly increased the number of AMRO people infected with COVID-19 and deaths in the first few months of the COVID-19 pandemic period. Therefore, the LTO is indeed not an important factor in influencing the number of AMRO people infected with COVID-19 in the first year of the COVID-19 pandemic period.

##### LTO and the WHO South East Asia (SEARO) Region

LTO only had a positive significant correlation with the CD-PM on 13 February 2021 and 20 February 2021. The SEARO region has a relative higher LTO mean value (47.40) for the six WHO regions, and SEARO people pay more attention to the future, believing that the traditions of the past will change with the times, and observing things from a dynamic point of view. However, the higher LTO score significantly increases, but not decreases, the number of deaths of people from the SEARO region. Meanwhile, the results show that the LTO score is indeed not an important factor in influencing the number of SEARO people infected with COVID-19 in the first year of the COVID-19 pandemic period.

##### LTO and the WHO Eastern Mediterranean (EMRO) Region

LTO only had a positive significant correlation with the NC-7A and NC on 22 June 2020. The EMRO region has a relative low LTO mean value (22.90) for the six WHO regions, and EMRO people pay more attention to current interests and pleasures; they hope to see results in a short time, and quick success is more urgent and cannot be delayed. However, the higher LTO score only slightly increased the number of EMRO people infected with COVID-19 in the fourth month of the COVID-19 pandemic period. Therefore, the LTO is indeed not an important factor in influencing the number of EMRO people infected with COVID-19 in the first year of the COVID-19 pandemic period.

##### LTO and the WHO Africa (AFRO) and Western Pacific (WPRO) Regions

LTO had no significant correlation with any of the CC, CC-PM, NC-7A, NC, CD, CD-PM, ND-7A, and ND throughout the first year of the COVID-19 pandemic period.

The AFRO region has a relatively low LTO mean value (23.83); therefore, AFRO people pay more attention to current interests and pleasures; they hope to see results in a short time, and quick success is more urgent and cannot be delayed. In contrast, the WPRO region has the highest LTO mean value (61.82); therefore, WPRO people look to the future more, believing that the traditions of the past will change with the times, and observing things from a dynamic point of view. The results show that LTO score is indeed not an important factor in influencing the number of people from these two regions infected with COVID-19 in the first year of the COVID-19 pandemic period.

#### 4.2.6. The Indulgence/Restraint (IVR) Correlation with COVID-19 Data (Figure A7, Figure A10, Figure A11 and Figure A17)

During the period from 22 February 2020 to 20 February 2021, IVR only had a positive significant correlation with the ND on 21 January 2021.

These results show that a lower IVR score increases the global death toll because of COVID-19 in the penultimate month of the first COVID-19 pandemic year. Indulgence represents the basic normal desire to enjoy the pleasures of life and allow unrestrained satisfaction; therefore, the results of this study show that the higher IVR score does not increase the global number of people infected with COVID-19. Meanwhile, the differences in each WHO region due to IVR scores will be discussed in the next section of this study.

##### IVR and the WHO Europe (EURO) Region

IVR only had a positive significant correlation with the CC-PM on 22 March 2020, 22 May 2020, and 22 June 2020. Meanwhile, IVR only had a positive significant correlation with the CD-PM from 22 May 2020 to 22 August 2020.

Although EURO has a relatively low IVR mean value (39.11), the higher IVR score increases some of the COVID-19 data of the EURO people in the beginning half of the first year of the COVID-19 pandemic period. Therefore, the IVR score is not an important factor in influencing the number of EURO people infected with COVID-19 in the first year of the COVID-19 pandemic period.

##### IVR and the WHO South East Asia (SEARO) Region

IVR only had a negative significant correlation with CC-PM on 22 July 2020. Although SEARO has a relatively low IVR mean value (32.25), the higher IVR score still increases some of the COVID-19 data from the SEARO people in the fifth month of the COVID-19 pandemic period. Therefore, the IVR score is not an important factor in influencing the number of SEARO people infected with COVID-19 in the first year of the COVID-19 pandemic period.

##### IVR and the WHO Africa (AFRO), Americas (AMRO), Western Pacific (WPRO), and Eastern Mediterranean (EMRO) Regions

IVR had no significant correlation with any of the CC, CC-PM, NC-7A, NC, CD, CD-PM, ND-7A, and ND throughout the first year of the COVID-19 pandemic period.

On the one hand, the AMRO region has the highest IVR mean value (69.94), the AFRO region has a relatively high IVR mean value (54.45), and the WPRO region has a relatively high IVR mean value (44.27); on the other hand, the EMRO region has the lowest IVR mean value (26.67). Meanwhile, the IVR score does not impact any of the COVID-19 data in these four WHO regions.

The results show that the societies either emphasize the basic normal desire to enjoy the pleasures of life and allow unrestrained satisfaction (high IVR score), or control the enjoyment of life and manage with strict social norms (low IVR score); the IVR score is indeed not an important factor in influencing the COVID-19 data of these four WHO regions in the first year of the COVID-19 pandemic period.

#### 4.2.7. Summary Discussion of Hofstede’s Six Dimensions and COVID-19 Data (See Figure A8, Figure A9 and Figure A10)

(1) Global: IDV has the highest correlation with COVID-19 data (all eight COVID-19 data in 12 months), PDI has the second highest correlation with COVID-19 data (all eight COVID-19 data in seven months), then followed by UAI (CC-PM and DC-PM in nine months), MAS (ND-7A and ND in three months), and LTO (CC, CC-PM and NC in five 5 months). Finally, IVR only has an ND correlation with COVID-19 data in one month.

(2) EURO region: IDV has the highest correlation with COVID-19 data (all eight COVID-19 data in 13 months), PDI has the second highest correlation with COVID-19 data (CC-PM, ND-7A, and ND in five months), then followed by IVR (CC-PM and DC-PM in five months), UAI (ND-7A and ND in one month), and LTO (CC-PM in one month). Finally, MAS has no correlation with any of the COVID-19 data.

(3) AFRO region: IDV has the highest correlation with COVID-19 data (all eight COVID-19 data in 12 months), PDI has the second highest correlation with COVID-19 data (all eight COVID-19 data in ten months), then followed by MAS (NC-7A, NC, and ND in five months), and UAI (CC-PM, CD, CD-PM, ND-7A, and ND in three months). Finally, LTO and IVR have no correlation with any of the COVID-19 data.

(4) AMRO region: IDV has the highest correlation with COVID-19 data (all eight COVID-19 data in 13 months), then followed by PDI (CC, NC-7A, NC, and CD-PM in two months), and LTO (NC-7A, NC, ND-7A, and ND in four months). Finally, MAS, UAI, and IVR have no correlation with any of the COVID-19 data.

(5) WPRO region: MAS has the highest correlation with COVID-19 data (NC-7A, NC, CD, ND-7A, and ND in six months), and UAI (NC-7A, NC, ND-7A, and ND in three months). Finally, PDI, IDV, MAS, LTO, and IVR have no correlation with any of the COVID-19 data.

(6) SEARO region: MAS has the highest correlation with COVID-19 data (CC-PM, CD-PM, ND-7A, and ND in six months), then followed by LTO (CD-PM in one month), UAI (CC-PM in one month), and IVR (CC-PM in one month). Finally, PDI and IDV have no correlation with any of the COVID-19 data.

(7) EMRO region: PDI has the highest correlation with COVID-19 data (CC-PM, NC-7A, NC, CD, ND-7A, and ND in eight months), then followed by IDV (NC-7A, NC, and CD-PM in three months), LTO (NC-7A and NC in one month), and MAS (NC in one month). Finally, UAI and IVR have no correlation with any of the COVID-19 data.

## 5. Contributions

### 5.1. For Academic

Although many scholars [4,10,13,24,27] have studied the correlation between Hofstede’s national culture and COVID-19, some scholars only focus on the Individualism and Uncertainty Avoidance dimensions, and some scholars [3,6,8,9,11,12,14,15,17] did not focus on the difference of Hofstede’s cultural dimensions to explore the gap between different WHO regions. In fact, this study found Hofstede’s six culture dimensions to significantly affect the COVID-19 data of different WHO regions in different ways, and it is a very critical insight in the academic field. Thus, the academic contributions will be discussed in the following sections.

Firstly, IDV is the most significant factor; the higher IDV score not only increases all of the global COVID-19 data, but also increases the EURO, AFRO, and AMRO people, and increases the CD-PM of the EMRO people. Therefore, in a society of emphasizing individualism, their people do increase their chances of becoming infected with COVID-19. However, the higher IDV decreases the NC-7A and NC of the EMRO people, and the result is probably because there are many countries in this region still in local wars. Meanwhile, the results are consistent with the studies of [4,10], and all of the global WHO data indeed prove the results of previous studies to be correct.

Secondly, PDI is the second important factor; the higher PDI score decreases all of the global and AFRO COVID-19 data, and decreases some of the COVID-19 data of the AMRO people. Thus, a society with higher power distance indeed decreases their number of people infected with COVID-19. Meanwhile, although PDI decreases some COVID-19 data of the EURO and EMRO people, higher PDI increases the ND-7A and ND in EURO, and CC-PM in EMRO. The result is probably because the medical systems of the EURO region were not capable of curing many COVID-19-infected people in the first few months of the first COVID-19 pandemic year, and people could not respect social distancing or the isolation strategies because of local wars in some EMRO countries. The PDI dimension result of this study is a critical new insight.

Thirdly, higher MAS is the third important factor; the higher MAS increases some COVID-19 data of the WPRO people, and increases some COVID-19 data of the AFRO, SEARO, and global people. Thus, there is only an impact on one COVID-19 datum of the EMRO people. Therefore, in a society of emphasizing masculinity, their people indeed increase the chances of being infected with COVID-19, but the higher MAS influence is relatively lower than the IDV and PDI. The result of the MAS dimension of this study also is an important new insight.

Fourthly, UAI is the fourth factor; the higher UAI only increases some of the COVID-19 cases of global, WPRO, AFRO, and EURO people. Thus, there is only an impact on one COVID-19 datum of the SEARO people. In addition, UAI is not an important factor in increasing the number of people infected with COVID-19; however, the result still proved that global WHO data is indeed consistent with previous studies [13,24,27].

Fifthly, LTO is the fifth factor; the higher LTO only increases some COVID-19 data of global, AMRO, and EMRO people, and only increases one COVID-19 datum of the EURO and SEARO people. Meanwhile, LTO is not an important factor in increasing the number of people infected with COVID-19, but the result of the LTO still is worthy of future reference for scholars. The result of the LTO dimension of this study is another new insight.

Sixthly, IVR is the last factor; the higher IVR increases some COVID-19 data of EURO people. In contrast, it decreases the COVID-19 factor of the global and SEARO people. IVR is a factor to be ignored in this study. However, the two opposite results indeed need to be explored in the future. The result of the IVR dimension of this study is also a new insight.

Finally, the AFRO region is the most important region where COVID-19 data has a higher correlation with PDI and IDV than MAS and UAI, followed by EURO and AMRO, where COVID-19 data have a higher correlation with PDI than IDV, IVR, UAI, and LTO in the EURO region, and than IDV and LTO in the AMRO region. Then, the EMRO region follows the above three regions where COVID-19 data have a higher correlation with PDI than IDV, LTO, MAS. Finally, WPRO and SEARO COVID-19 data have a higher correlation with MAS than UAI in the WPRO region, and UAI and LTO in the SEARO region.

### 5.2. For Practice

Based on the results of this study, governments and people in different cultural regions can learn how to more effectively deal with the spread of infectious diseases in the future. Thus, in the following section the contributions for practice will be discussed.

Firstly, a higher IDV score indeed significantly increases many COVID-19 data in many WHO regions (AFRO, EURO, AMRO, and EMRO), and the government of an individualistic society can request the temporary sacrifice of personal freedom to protect the health and safety of the public.

Secondly, a higher PDI score also significantly decreases many COVID-19 data in many WHO regions (AFRO, EURO, AMRO, and EMRO), and a society of lower power distance can be requested to temporarily abide by the government’s policy against infectious diseases to protect the health and safety of the public.

Thirdly, a higher MAS significantly increases some or a few COVID-19 data in many regions (EURO, WPRO, SEARO, and EMRO), and a government in a high-masculinity society can propagate the characteristics of masculinity among the people to protect the vulnerable and cooperate with the government’s policy of reducing infectious diseases to protect the health and safety of the public.

Fourthly, a higher UAI only slightly increases a few COVID-19 data in many regions (EURO, AFRO, WPRO, and SEARO), and a society with a higher UAI score expects a reduction in uncertainty which can calm people’s minds. The UAI is not an important factor in increasing COVID-19 data; however, governments still can use the UAI characteristic to reduce the uncertainty in everyone’s mind, and people should cooperate with the government’s policy of reducing infectious diseases to ensure the health and safety of the public. 

Fifthly, a higher LTO only slightly increases a few COVID-19 data in many regions (EURO, AMRO, SEARO, and EMRO), and people in a high LTO society like to look to the future, believing that the traditions of the past will change with the times, and observing things from a dynamic point of view, so there will be room for everything. Although the LTO is not an important factor in increasing COVID-19 data, governments still have a responsibility to request that people try their best to cooperate with their policies for reducing infectious diseases to ensure the health and safety of the public.

Sixthly, a higher IVR only slightly increases two COVID-19 data in the EURO region, and decreases one COVID-19 datum in the SEARO region. When people wish to enjoy the pleasures of life, have indulgent thoughts, and become a society of unrestrained satisfaction, the higher IVR of course increases COVID-19; however, why this result only exists in the EURO region, but has a contrasting result in SEARO, is a question to explore in the future. Even though IVR is a negligible factor in the spread of COVID-19 in society, governments should also pay attention to the possible harm of IVR in the spread of infectious diseases in the future, and propose effective strategies of prevention and control.

Seventhly, the governments of the AFRO region should pay special attention to the correlation between PDI, IDV, and infectious diseases, and dealing with these two aspects can achieve effective results in inhibiting it. Meanwhile, the governments of EURO and AMRO should pay special attention to the correlation between PDI and infectious diseases. From then on, effective results can be achieved to curb it. Then, even though the correlation between EMRO’s COVID-19 data and Hofstede’s cultural dimensions is not great, governments should still pay attention to the possible correlation between PDI and infectious diseases as an effective containment strategy. Finally, although the correlation between COVID-19 and Hofstede’s cultural dimensions in WPRO and SEARO are so small that it can be ignored, governments should also pay attention to the possible correlation between MAS and infectious diseases as an effective containment strategy.

## 6. Conclusions

Global data have been collected in this study to explore the correlations between Hofstede’s six cultural dimensions (PDI, IDV, MAS, UAI, LTO, and IVR) and COVID-19 data in the first year of the COVID-19 pandemic period. It was found that Hofstede’s six culture dimensions have significantly affected the COVID-19 data of different WHO regions in different ways. In which, four of Hofstede’s dimensions, PDI, MAS, LTO, and IVR, are worthy of further study by scholars. The results of this study will also be of greater help to global governments and medical institutions in formulating policies to suppress infectious diseases in the future.

COVID-19 data were collected from 240 countries in this study; however, Hofstede only collected data from 117 countries. Therefore, COVID-19 data from 123 countries have to be ignored in the current study, and this is the first limitation of this study. Meanwhile, because some countries are still at war, some countries have no sound medical systems, and PCR tests might be not widely available in the first year; all of these factors will influence the WHO in the collection of COVID-19 data, and then the results of this study also have bias too. This is the second limitation of this study. In addition, some of Hofstede’s cultural dimensions (e.g., PDI has two opposite (positive/negative) significant correlations with COVID-19 data in the same/different WHO regions at the same time (e.g., PDI in EURO (+/−) and EMRO (+/−); IDV in EMRO (+/−); LTO in EURO (−) is different with AMRO (+), SEARO (+), and EMRO (+); IVR in SEARO (−) is different with EURO (+)), and the conflict and opposing results are worthy of further study by scholars. Because of the isolation policy in the first year of the COVID-19 pandemic period, this study had no chance to collect more detailed data for each country to explore the reason. This is the third limitation of this study. Meanwhile, the classification of the WHO regions is for the convenience of the WHO management; thus, many different cultural countries/areas exist in the same WHO region, and the highest or the lowest score of Hofstede’s six cultural dimensions exist in the same WHO region (e.g., 11 and 100 of PDI and five and 100 of MAS in EURO; six and 91 of IDV in AMRO). This is the fourth limitation of this study. For this reason, it is necessary to adopt a more appropriate classification to classify countries/areas in future study. Finally, IT is indeed a critical factor in inhibiting the spread of COVID-19, but the relationship between IT and COVID-19 data was not analyzed in this study, and this is the fifth limitation of this study.

## Figures and Tables

**Table 1 healthcare-11-02258-t001:** Descriptions of the variables of this study.

Variable	Abbreviation	References
Cumulative_cases	CC	WHO [31]; GCDL [32]; HCHC [33]
Cases—cumulative total per 1 million population	CC-PM	
Cases—newly reported in last 7 days (average)	NC-7A	
New_cases	NC	
Cumulative_deaths	CD	
Deaths—cumulative total per 1 million population	CD-PM	
Deaths—newly reported in last 7 days (average)	ND-7A	
New_deaths	ND	
Power Distance	PDI	Hofstede [19]
Individualism/Collectivism	IDV	
Masculinity/Femininity	MAS	
Uncertainty Avoidance	UAI	
Long-Term/Short-Term Orientation	LTO	
Indulgence/Restraint	IVR	

## Data Availability

Data are available upon request to the contact author.

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
