# Peer review of "The Correlation between Hofstede’s Cultural Dimensions and COVID-19 Data in the Early Stage of the COVID-19 Pandemic Period"

_healthcare, 2023, doi:10.3390/healthcare11162258_

Round 1

Reviewer 1 Report

Dear authors,

The threat of COVID-19 to humanity is basically over, and it may become seasonal influenza. However, from the end of 2019 to the present, the impact of the pandemic on human society has been far-reaching, and many people have lost their lives as a result. Continuous research on this topic is valued by value.

Judging from the content of the manuscript, the authors spent a lot of time collecting very detailed data and materials. The research methodology used in this study is reasonable and feasible. The purpose of the study was relatively clear. 

In order to better improve the readability and quality of the manuscript, the following comments or suggestions are for your reference.

1)     There is a large amount of data in the manuscript, can you explain how it was collected?

2)     Why did authors choose these 6 dimensions?

3)     Chapter 4 is huge and detailed, which is very worthy of recognition! But if you can put it together into a concise chart, that's great.

4)     Chapter VI needs to be improved. First, part of it is still a discussion of data, and it needs to be further summarized. In addition, there were 4 limitations in the study, why were they not discovered at the beginning of the study? Do these limitations have a significant impact on the confidence of the findings? These need to be explained by the authors.

5)     Perhaps the authors think that there is too much data, and if inserted in the paragraph, it may affect the efficiency of reading. But even if these data are placed in an appendix, they should be clearly accessible to the reader. Current fonts are too small to be recognized effectively.

Author Response

Response to Reviewer 1 Comments

The threat of COVID-19 to humanity is basically over, and it may become seasonal influenza. However, from the end of 2019 to the present, the impact of the pandemic on human society has been far-reaching, and many people have lost their lives as a result. Continuous research on this topic is valued by value.

Judging from the content of the manuscript, the authors spent a lot of time collecting very detailed data and materials. The research methodology used in this study is reasonable and feasible. The purpose of the study was relatively clear.

In order to better improve the readability and quality of the manuscript, the following comments or suggestions are for your reference.

1) There is a large amount of data in the manuscript, can you explain how it was collected?

Authors' response: Thank you for your comment.

The data of this study was collected through related websites per month from 2020/02/22 to 2021/02/20. These websites have described in section 3.1 (p. 7, line 330-341), and hopefully the description is now acceptable.

2) Why did authors choose these 6 dimensions?

Authors' response: Thank you for your comment.

These six dimensions are based on the Nation Cultural Theory of Hofstede (2020), and described in section 2.1 (pp. 3-7, line 115-315). Hopefully the description is now acceptable.

3) Chapter 4 is huge and detailed, which is very worthy of recognition! But if you can put it together into a concise chart, that's great.

Authors' response: Thank you for your comment.

Because the data of this study is huge and detailed, therefore, it cannot be put together into a concise chart. Meanwhile, the clearer and readable appendices of Table A2 and Table A3 are provided in the new version. Hopefully the description is now acceptable.

4) Chapter VI needs to be improved. First, part of it is still a discussion of data, and it needs to be further summarized. In addition, there were 4 limitations in the study, why were they not discovered at the beginning of the study? Do these limitations have a significant impact on the confidence of the findings? These need to be explained by the authors.

Authors' response: Thank you for your comment.

  1. The first paragraph of the section 6. Conclusion has been summarized in the new version (p. 21, line 1002-1009), and hopefully the description is now acceptable.
  2. The five limitations has been added in the last paragraph of section 1. Introduction (p. 2, line 78-87), and hopefully the description is now acceptable.
  3. Although there are five limitations of this study, however, the confidence of the findings will not be impacted. Hopefully the description is now acceptable.

5) Perhaps the authors think that there is too much data, and if inserted in the paragraph, it may affect the efficiency of reading. But even if these data are placed in an appendix, they should be clearly accessible to the reader. Current fonts are too small to be recognized effectively.

Authors' response: Thank you for your comment.

The clearer and readable appendices of Table A2 and Table A3 are provided in the new version. Hopefully the description is now acceptable.

Reviewer 2 Report

The study brings a new perspective to assess the epidemiological risk and the potential public health approach required to control future epidemics, considering the population cultural profile. Although the analysis used country-level cultural scores aggregated by regions, the results seem to reflect what is known from the social, economic, and cultural aspects of each WHO Region. The possible public health use of the findings should be obviously adapted to country and local contexts.  It would be interesting to further discuss the studies required to validate the results giving more specificity to it and explain how this approach can assist the design of public health approaches for controlling future epidemics.

Author Response

Response to Reviewer 2 Comments

The study brings a new perspective to assess the epidemiological risk and the potential public health approach required to control future epidemics, considering the population cultural profile. Although the analysis used country-level cultural scores aggregated by regions, the results seem to reflect what is known from the social, economic, and cultural aspects of each WHO Region. The possible public health use of the findings should be obviously adapted to country and local contexts. It would be interesting to further discuss the studies required to validate the results giving more specificity to it and explain how this approach can assist the design of public health approaches for controlling future epidemics.

Authors' response: Thank you for your comment.

Reviewer 3 Report

The paper presented for publication in healthcare journal must be improved in the empirical section because now it is impossible to analyze in a clear form. For this:

1.- The authors must think about other form to present the information in the appendix table A2 because now they are illegible.

2.- Is there more information? Maybe it is interesting to consider information in several point more separated in the time that 17 consecutive observations that it makes the information illegible. Moreover, the authors could consider a longer time period.

3.- The results section must be written in a clearer form because now it is difficult to understand, and it is not necessary consider one sub epigraph for each relationships.

4.- Have you think about the possible problem of the different size of the group? Is it possible to divide Europe in order to consider a more homogeneous group size?

5.- Finally, you must consider the possibility to stablish a regression model to consider the variable more important in each case or considering a categorical variable with the WHO region and use an ANOVA analysis. Thus the analysis will be more complete.

Author Response

Response to Reviewer 3 Comments

The paper presented for publication in healthcare journal must be improved in the empirical section because now it is impossible to analyze in a clear form. For this:

1.-The authors must think about other form to present the information in the appendix table A2 because now they are illegible.

Authors' response: Thank you for your comment.

The clearer and readable appendices of Table A2 and Table A3 are provided in the new version. Hopefully the description is now acceptable.

2.-Is there more information? Maybe it is interesting to consider information in several point more separated in the time that 17 consecutive observations that it makes the information illegible. Moreover, the authors could consider a longer time period.

Authors' response: Thank you for your comment.

The data of 17 times can show the spread trend of COVID-19 in different WHO regions, so it is necessary to collect, and hopefully the description is now acceptable.

3.-The results section must be written in a clearer form because now it is difficult to understand, and it is not necessary consider one sub epigraph for each relationship.

Authors' response: Thank you for your comment.

Each relationship is clearly written in order to let readers understand the degree of correlation between different cultural dimensions, different time of COVID-19 data, the spread of COVID-19 in different WHO regions, and the importance of different cultural dimensions to the different WHO regions.

4.-Have you think about the possible problem of the different size of the group? Is it possible to divide Europe in order to consider a more homogeneous group size?

Authors' response: Thank you for your comment.

Because the data of this study collected from WHO, therefore, this study has adopted the six WHO regions to classify these data. The weakness of the classification has been found in the end of this study. In light of this, this study has added it to be the fourth limitation of this study, and it will be improved in the future study. Hopefully the description is now acceptable.

5.-Finally, you must consider the possibility to stablish a regression model to consider the variable more important in each case or considering a categorical variable with the WHO region and use an ANOVA analysis. Thus, the analysis will be more complete.

Authors' response: Thank you for your comment.

In order to understand the trend of correlation between COVID-19 at different time points in different regions of WHO in different cultural dimensions, ANOVA cannot achieve the purpose of this study, so ANOVA analysis was not used in this study.

Round 2

Reviewer 3 Report

The authors have answered my suggestions.